# Equilibrium and Non-Equilibrium Lattice Dynamics of Anharmonic Systems

**DOI:** 10.3390/e24111585

**Published:** 2022-11-01

**Authors:** Keivan Esfarjani, Yuan Liang

**Affiliations:** 1Department of Mechanical and Aerospace Engineering, University of Virginia, Charlottesville, VA 22904, USA; 2Department of Materials Science and Engineering, University of Virginia, Charlottesville, VA 22904, USA; 3Department of Physics, University of Virginia, Charlottesville, VA 22904, USA

**Keywords:** nanoscale thermal transport, anharmonicity, phonons, heat current, transmission, thermal conductance

## Abstract

In this review, motivated by the recent interest in high-temperature materials, we review our recent progress in theories of lattice dynamics in and out of equilibrium. To investigate thermodynamic properties of anharmonic crystals, the self-consistent phonon theory was developed, mainly in the 1960s, for rare gas atoms and quantum crystals. We have extended this theory to investigate the properties of the equilibrium state of a crystal, including its unit cell shape and size, atomic positions and lattice dynamical properties. Using the equation-of-motion method combined with the fluctuation–dissipation theorem and the Donsker–Furutsu–Novikov (DFN) theorem, this approach was also extended to investigate the non-equilibrium case where there is heat flow across a junction or an interface. The formalism is a classical one and therefore valid at high temperatures.

## 1. Introduction

There has recently been intense research on developing high-temperature materials that can operate at temperatures of around 1500 K or above. These materials can potentially be used in jet engines, petrochemical and material processing, and power plants based on concentrated solar power. Another application would be in hypersonic vehicles, for which the surface temperature can exceed 2000 ${}^{\circ }$C. These materials are typically borides or carbides made of refractory elements along with their high-entropy alloy combinations [1,2]. At such high temperatures, the atomic vibrational amplitude, being proportional to ${\left(kT/m\omega^{2}\right)}^{1/2}$, grows beyond the region where harmonic approximation is valid, and therefore, more sophisticated models are required to describe such dynamics. A successful yet relatively simple model that can handle anharmonic systems is the self-consistent phonon (SCP) theory originally proposed by Born and Hooton [3]. This theory was then extended and further developed by Boccara and Sarma [4] using a variational approach based on free-energy minimization in order to study displacive phase transitions. Later, Choquard used the cluster expansion theory [5] and Ranninger used many-body diagrammatic theory [6] to derive similar equations. The first complete SCP theory was, however, initially worked out as a mean-field theory by Gillis et al. in 1968 [7]. It was then completed to second-order by Werthamer in 1970 [8].

In the first section of this review, we go over this theory, its variational formulation and its modern implementations based on first-principles Density Functional Theory (DFT) calculations. The second part of this review deals with the non-equilibrium case where an anharmonic device is connected to multiple probes or thermostats with different temperatures. We then derive, based on the equation of motion of atoms in the device, an expression for the thermal current and entropy generation in the device.

## 2. The Self-Consistent Phonon Theory

Following the variational formulation first proposed by Boccara and Sarma [4], we ask ourselves: For a system at a given temperature, what is the “best” harmonic Hamiltonian that describes the thermodynamics of this system (the “best” being defined as the one that minimizes a variational free energy)? According to Bogoliubov’s inequality [9], if the Hamiltonian is a sum of a solvable Hamiltonian $H_{0}=T+V_{0}$ and a remaining potential energy term $V-V_{0}$, then its exact free energy satisfies the following inequality:(1)$$F\leq F_{trial}=F_{0}+{\langle V-V_{0}\rangle }_{0}$$

The average in this inequality is with respect to the density matrix associated with the solvable Hamiltonian $H_{0}$ [9]:(2)$${\langle A\rangle }_{0}=Tr\rho_{0}A;\;\;\rho_{0}=e^{-H_{0}/kT}/Z;\;\;Z=\mathrm{Tr}\;e^{-H_{0}/kT};\;F_{0}=-kT\ln Z$$

In the solvable Hamiltonian $H_{0}$, which, in our case, is a harmonic Hamiltonian, the force constants between pairs of atoms are free parameters that can be determined from the above minimization once the potential energy *V* is known and its average can be calculated. In what follows, we assume that the potential energy *V* is a polynomial in powers of atomic displacements about their equilibrium positions, and we truncate this expansion at the fourth-order:(3)$$\begin{array}{cc} H=T+V & =\sum_{i}\frac{p_{i}^{2}}{2m_{i}}+\sum_{ij}\;\frac{1}{2!}\phi_{ij}\;u_{i}\;u_{j}+\sum_{ijk}\;\frac{1}{3!}\psi_{ijk}\;u_{i}\;u_{j}\;u_{k}+\sum_{ijkl}\;\frac{1}{4!}\chi_{ijkl}\;u_{i}\;u_{j}\;u_{k}\;u_{l} \end{array}$$
where i,j,k,l refer to atoms, and $u_{i}$, $p_{i}$, and $m_{i}$ refer to the displacement, momentum, and mass, respectively, of atom *i*. The Greek letters ϕ,ψ,χ are, respectively, the second, third and fourth derivatives of the potential energy evaluated at the equilibrium positions. They are also called the force constants, and are harmonic in the case of ϕ and anharmonic for ψ and χ. We have omitted the Cartesian indices for brevity. In case there are multiple atoms in a unit cell, the label *i* can more generally be extended to include the labels τ of the atoms in the cell labeled by *R*, as well as the Cartesian component of *u*. In other words, i=(Rτα). In order to include structural phase transitions and thermal expansion, we introduce two additional sets of variational parameters: the strain tensor η and $u_{i}^{0}$, which is the internal relaxation of atoms beyond the strain effect (captured by $\eta R_{i}$) due to changes in the unit cell shape or volume. Thus, the general displacement of atom *i* from its reference position as a result of cell deformation can be written as: $u_{i}\left(t\right)=\eta R_{i}+u_{i}^{0}+\left(1+\eta \right)y_{i}\left(t\right)=s_{i}+\left(1+\eta \right)y_{i}\left(t\right)$, where $y_{i}\left(t\right)$ is the dynamical displacement of atom *i* about its equilibrium, so that by definition, ${\langle y_{i}\rangle }_{0}=0$. The vector $R_{i}$ refers to the lattice translation vector of the cell containing *i*. Further, $s_{i}=\langle u_{i}\rangle =\eta R_{i}+u_{i}^{0}$ is the static displacement of atom *i* due to strain and internal relaxations, and is introduced for brevity of notations. The coefficients of this expansion can be obtained from a zero-temperature DFT calculation, for instance, as in [10,11].

The trial harmonic potential containing the variational parameters $K_{ij}$ is defined as:(4)$$\begin{array}{cc} V_{0} & =\sum_{ij}\;\frac{1}{2!}K_{ij}\;y_{i}\;y_{j} \end{array}$$

For the trial harmonic Hamiltonian, the free energy can be calculated analytically, as it is the sum of free energies associate with each harmonic mode:(5)$$F_{0}=kT\sum_{k\lambda }\ln\left[2\sinh\frac{\beta \hbar \omega_{k\lambda }}{2}\right]$$
where $\beta =1/k_{B}T$ and $\omega_{k\lambda }$ are, respectively, the harmonic frequency of mode λ and the wave-vector *k* obtained from diagonalizing the dynamical matrix associated with the trial force constants: $D_{\tau \tau^{\prime }}\left(k\right)=\sum_{R}\frac{K_{\tau ,R+\tau^{\prime }}}{\sqrt{m_{\tau }m_{\tau^{\prime }}}}\;e^{ik\cdot (R+\tau^{\prime }-\tau )}$, where *k* refers to a *k* vector in the first Brillouin zone, and the indices τ refer to atoms in the primitive cell. Note that the existence of the term $\tau -\tau^{\prime }$ in the exponent does not change the eigenvalues and only causes a phase shift in the eigenfunctions. The matrix *D*, being Hermitian, has real eigenvalues denoted by $\omega_{k\lambda }^{2}$ and eigenvectors $e_{\tau \alpha ,\lambda }\left(k\right)$, where α is the Cartesian coordinate and λ refers to the vibrational mode (τα can be understood as the line index and λ as the column index of the unitary eigenvector matrix *e* for each *k*). We also need the thermal averages of the trial and anharmonic potentials, $V_{0}$ and *V*, respectively. In terms of the eigenvalues and eigenvectors of the above dynamical matrix and the phonon creation and annihilation operators *a* and $a^{†}$, respectively, the displacements *y* can be written as [12]
(6)$$y_{R\tau \alpha }=\sum_{k\lambda }\sqrt{\frac{\hbar }{2Nm_{\tau }\omega_{k\lambda }}}\;e_{\tau \alpha ,\lambda }\left(k\right)\;\left(a_{k\lambda }+a_{-k\lambda }^{†}\right)\;e^{i\vec{k}\cdot \left(\vec{R}+\vec{\tau }\right)}$$
so that for the thermal average of displacement autocorrelations, we have
(7)$$\begin{array}{c} \langle y_{R\tau \alpha }\;y_{R^{\prime }\tau^{\prime }\beta }\rangle =\frac{\hbar }{2N}\sum_{k,\lambda }\frac{\left(2n_{k\lambda }+1\right)}{\omega_{k\lambda }}\;\frac{e_{\tau \alpha ,\lambda }\left(k\right)\;e_{\tau^{\prime }\beta ,\lambda }\left(-k\right)}{\sqrt{m_{\tau }m_{\tau^{\prime }}}}e^{-i\vec{k}\cdot \left(\vec{R^{\prime }}+\vec{\tau^{\prime }}-\vec{R}-\vec{\tau }\right)} \end{array}$$
where $\langle a_{k\lambda }^{†}a_{k^{\prime }\lambda^{\prime }}\rangle =n_{k\lambda }\delta_{\lambda ,\lambda^{\prime }}\delta_{k,k^{\prime }};\;\langle a_{k^{\prime }\lambda^{\prime }}a_{k\lambda }^{†}\rangle =\left(n_{k\lambda }+1\right)\delta_{\lambda ,\lambda^{\prime }}\delta_{k,k^{\prime }}$ were used. Using the identity $2n_{k\lambda }+1=\coth\frac{\beta \hbar \omega_{k\lambda }}{2}$, the equilibrium average of the harmonic potential can be written as:(8)$$\langle V_{0}\rangle =\sum_{ij}\;\frac{1}{2!}K_{ij}\;\langle y_{i}y_{j}\rangle =\sum_{k\lambda }\frac{\hbar \omega_{k\lambda }}{4}\coth\frac{\beta \hbar \omega_{k\lambda }}{2}$$

We see, therefore, that the trial free energy and harmonic potential are both only functions of the harmonic frequencies $\omega_{k\lambda }$, and thus $K_{ij}$, while the average of the anharmonic potential *V* depends also on η and $u_{i}^{0}$. As for the model anharmonic potential expressed in Equation (Equation 3), since the variable *y* has a Gaussian distribution, the average of its odd powers is zero, and thus, the thermal average of the actual potential 〈V〉 can be presented in Equation (Equation 9) as:(9)$$\begin{array}{cc} \langle V(C,s)\rangle \; & =\frac{1}{2}\sum_{ij}\;\phi_{ij}\;C_{ij}+\frac{1}{2}\sum_{ijk}\psi_{ijk}\;C_{ij}s_{k}+\frac{1}{8}\sum_{ijkl}\chi_{ijkl}\;C_{ij}C_{kl} \end{array}$$
where for the sake of simplicity of notation, all the Cartesian indices have been omitted, and we used $C_{ij}=\langle u_{i}\;u_{j}\rangle =s_{i}s_{j}+{\left(1+\eta \right)}^{2}\langle y_{i}\;y_{j}\rangle$. We also used the invariance of the force constants under permutation of indices. Further, note that in *C*, each term (1+η) is always contracted with one $y_{i}$.

### 2.1. Determination of the State of Thermal Equilibrium at a Given T

Having the equilibrium average of the anharmonic potential, we can proceed to minimize the variational or trial free energy $F_{trial}=F_{0}\left(K\right)+\langle V\left(C,s\right)-V_{0}\left(K\right)\rangle$ in an iterative manner by setting its gradients with respect to different variational parameters to zero and solving the resulting set of non-linear equations in $\left(u^{0},\eta ,K\right)$. If we think of $K_{ij}$ and $C_{ij}=\langle u_{i}\;u_{j}\rangle =s_{i}s_{j}+{\left(1+\eta \right)}^{2}\langle y_{i}\;y_{j}\rangle$ as *independent* variables in order to avoid the use of the chain rule to express this dependence, minimization of the trial free energy $F_{trial}\left(u^{0},\eta ,K,C\right)$ leads to an additional, fourth non-linear equation. Thus, we have the following gradient formulas resulting from the minimization condition:(10)$$\begin{array}{ccc} \frac{\partial F_{trial}}{\partial u_{i}^{0}} & = & \frac{\partial \langle V\rangle }{\partial u_{i}^{0}}=\sum_{jk}\frac{\partial \langle V\rangle }{\partial C_{jk}}\frac{\partial C_{jk}}{\partial u_{i}^{0}}+\frac{\partial \langle V\rangle }{\partial s_{i}}=0 \end{array}$$(11)$$\begin{array}{ccc} \frac{\partial F_{trial}}{\partial \eta } & = & \sum_{ij}\frac{\partial \langle V\rangle }{\partial C_{ij}}\frac{\partial C_{ij}}{\partial \eta }+\sum_{i}\frac{\partial \langle V\rangle }{\partial s_{i}}\frac{\partial s_{i}}{\partial \eta }=0 \end{array}$$(12)$$\begin{array}{ccc} \frac{\partial F_{trial}}{\partial K_{ij}} & = & \frac{\partial F_{0}}{\partial K_{ij}}-\frac{\partial V_{0}}{\partial K_{ij}}=\frac{\partial F_{0}}{\partial K_{ij}}-\frac{1}{2}\langle y_{i}y_{j}\rangle =0 \end{array}$$(13)$$\begin{array}{ccc} \frac{\partial F_{trial}}{\partial C_{ij}} & = & \frac{\partial \langle V\rangle }{\partial C_{ij}}-\frac{\partial \langle V_{0}\rangle }{\partial C_{ij}}=\frac{\partial \langle V\rangle }{\partial C_{ij}}-\frac{1}{2}K_{ij}=0 \end{array}$$

The first two equations express the equilibrium conditions: no force on atoms and no stress in the unit cell. In the third Equation (Equation 12), since the anharmonic part 〈V〉 does not depend on *K*, we recover a general thermal property of harmonic Hamiltonians that leads, using the chain rule, to the expression of the displacement correlations in terms of the eigenvalues and eigenvectors of the dynamical matrix, as we have expressed in Equation (Equation 7). Finally, the fourth equation defines what the effective or optimal force constants *K* need to be in terms of the parameters of the potential energy *V*. Boccara and Sarma [4] showed that effective force constants are given by the thermal average of the second derivative of the potential energy: $K_{ij}={\langle \frac{\partial^{2}V}{\partial u_{i}\partial u_{j}}\rangle }_{0}$, which gives them a clear physical interpretation: The effective harmonic force constants are the *thermal average* of the second derivative of the potential, i.e., instead of evaluating the second derivative at $u_{i}=0$, it needs to be averaged over all values of the dynamical variables $u_{i}\left(t\right)$ weighted by their Boltzmann distribution probability.

Equation (Equation 13) provides a relationship between $C_{ij}$ and $K_{ij}$, allowing elimination of *K* as a variational parameter. From Equation (Equation 9), we have:(14)$$\frac{\partial \langle V\rangle }{\partial C_{ij}}=\frac{1}{2}\phi_{ij}+\frac{1}{2}\sum_{k}\psi_{ijk}s_{k}+\sum_{kl}\frac{1}{4}\chi_{ijkl}C_{kl}$$
so that Equation (Equation 13) reduces to the thermal-averaged second derivative:(15)$$\begin{array}{c} K_{ij}=\phi_{ij}+\sum_{k}\psi_{ijk}s_{k}+\sum_{kl}\frac{1}{2}\chi_{ijkl}C_{kl} \end{array}$$

Therefore, the set of non-linear equations to solve for $\left(u^{0},\eta ,K,C\right)$ are Equations (Equation 7), (Equation 10), (Equation 11) and (Equation 15). Note, this is a set of self-consistent equations, as the knowledge of $K_{ij}$ depends, through Equation (Equation 15), on $C_{ij}$ or $\langle y_{i}y_{j}\rangle$, which, in turn, depends on the eigenvalues and eigenvectors of *K* through Equation (Equation 7).

If one wants to impose an external pressure or a general stress tensor and find the equilibrium state under applied external stress $\sigma_{ext}$, the following thermodynamic potential $F\left(\eta \right)-\int d^{3}r\;\sigma_{\alpha \beta }^{ext}\;\eta_{\alpha \beta }$ needs to be minimized with respect to η.

Since we have considered *C* and *s* to be independent, we have:(16)$$\begin{array}{ccc} \frac{\partial \langle V\rangle }{\partial s_{i}} & = & \frac{1}{2}\sum_{jk}\psi_{ijk}\;C_{jk} \end{array}$$

For convenience, we also reproduce the derivatives of the quantities *s* and *C* with respect to $u^{0},\eta$, in full notation:(17)$$\begin{array}{ccc} \frac{\partial s_{R^{\prime }\tau^{\prime }\mu }}{\partial u_{R\tau \alpha }^{0}} & = & \delta_{RR^{\prime }}\;\delta_{\tau \tau^{\prime }}\;\delta_{\alpha \mu } \end{array}$$(18)$$\begin{array}{ccc} \frac{\partial s_{R\tau \mu }}{\partial \eta^{\alpha \beta }} & = & {\left(R+\tau \right)}_{\beta }\;\delta_{\alpha \mu }+{\left(R+\tau \right)}_{\alpha }\;\delta_{\beta \mu } \end{array}$$(19)$$\begin{array}{ccc} \frac{\partial C_{R\tau \mu ,R^{\prime }\tau^{\prime }\nu }}{\partial u_{R^{\prime \prime }\tau^{\prime \prime }\alpha }^{0}} & = & s_{R\tau \mu }\;\delta_{R^{\prime }+\tau^{\prime },R^{\prime \prime }+\tau^{\prime \prime }}\;\delta_{\alpha \nu }+s_{R^{\prime }\tau^{\prime }\nu }\;\delta_{R+\tau ,R^{\prime \prime }+\tau^{\prime \prime }}\;\delta_{\alpha \mu } \end{array}$$
(20)$$\begin{array}{ccc} \frac{\partial C_{R\tau \mu ,R^{\prime }\tau^{\prime }\nu }}{\partial \eta^{\alpha \beta }} & = & s_{R\tau \mu }\left[{\left(R^{\prime }+\tau^{\prime }\right)}_{\beta }\;\delta_{\alpha \nu }+{\left(R^{\prime }+\tau^{\prime }\right)}_{\alpha }\;\delta_{\beta \nu }\right]+s_{R^{\prime }\tau^{\prime }\nu }\left[{\left(R+\tau \right)}_{\beta }\;\delta_{\alpha \mu }+{\left(R+\tau \right)}_{\alpha }\;\delta_{\beta \mu }\right]\\ & + & {\left(I+\eta \right)}^{\mu \mu^{\prime }}\left(\langle y_{R\tau \mu^{\prime }}\;y_{R^{\prime }\tau^{\prime }\beta }\rangle \;\delta_{\alpha \nu }+\langle y_{R\tau \mu^{\prime }}\;y_{R^{\prime }\tau^{\prime }\alpha }\rangle \;\delta_{\beta \nu }\right) \end{array}$$

In conclusion, for the adopted model potential in Equation (Equation 3), the four non-linear equations to be solved self-consistently are Equations (Equation 7), (Equation 15), (Equation 21) and (Equation 22). The latter two are reproduced below, including all Cartesian indices for the sake of clarity and completeness:(21)$$\begin{array}{ccc} \frac{\partial F_{trial}}{\partial u_{R\tau \alpha }^{0}} & = & \sum_{R^{\prime }\tau^{\prime }\beta }K_{R\tau \alpha ,R^{\prime }\tau^{\prime }\beta }\;s_{R^{\prime }\tau^{\prime }\beta }+\frac{1}{2}\sum_{R^{\prime }\tau^{\prime }\beta R^{\prime \prime }\tau^{\prime \prime }\gamma }\psi_{R\tau \alpha ,R^{\prime }\tau^{\prime }\beta ,R^{\prime \prime }\tau^{\prime \prime }\gamma }\;C_{R^{\prime }\tau^{\prime }\beta ,R^{\prime \prime }\tau^{\prime \prime }\gamma }=0 \end{array}$$
(22)$$\begin{array}{ccc} \frac{\partial F_{trial}}{\partial \eta^{\alpha \beta }} & = & \sum_{R\tau \mu ,R^{\prime }\tau^{\prime }\nu }\left(K_{R\tau \mu ,R^{\prime }\tau^{\prime }\nu }\;s_{R^{\prime }\tau^{\prime }\nu }+\frac{1}{2}\psi_{R\tau \mu ,R^{\prime }\tau^{\prime }\nu ,R^{\prime \prime }\tau^{\prime \prime }\sigma }\;C_{R^{\prime }\tau^{\prime }\nu ,R^{\prime \prime }\tau^{\prime \prime }\sigma }\right)\;\left(R_{R\tau \beta }\;\delta_{\mu \alpha }+R_{R\tau \alpha }\;\delta_{\mu \beta }\right)\\ & + & K_{R\tau \mu ,R^{\prime }\tau^{\prime }\nu }\;{\left(1+\eta \right)}^{\mu \mu^{\prime }}\left(\delta_{\nu \beta }\langle y_{R\tau \mu^{\prime }}y_{R^{\prime }\tau^{\prime }\alpha }\rangle +\delta_{\nu \alpha }\langle y_{R\tau \mu^{\prime }}y_{R^{\prime }\tau^{\prime }\beta }\rangle \right)=0 \end{array}$$

The self-consistent procedure may be implemented as follows:

(0) Start from initial guesses for $\left(u^{0},\eta ,K_{ij}\right)$;

(1) Diagonalize the dynamical matrix obtained from *K*, and calculate $\langle y_{i}y_{j}\rangle$ from Equation (Equation 7);

(2) Use $\langle y_{i}y_{j}\rangle$ and η in Equation (Equation 21) to solve for $u^{0}$ contained in *s* and *C*;

(3) Use Equation (Equation 22) to solve for η from the knowledge of $u^{0}$, *K* and 〈yy〉;

(4) Use (s,C) to calculate the new *K* from Equation (Equation 15);

(5) Eventually mix the result with old *K*s to avoid numerical instabilities;

(6) Go to (1) until the process converges.

Note the non-linear Equations above, (Equation 21) and (Equation 22), are only quadratic in $u^{0}$ and η.

This procedure is not always straightforward, and care needs to be taken when solving numerically: for instance, the dynamical matrix obtained, *K*, in Step 1 may not be positive definite. Another subtlety is in solving Equation (Equation 16), which is a non-linear equation in *S* and may have multiple solutions. In this review, we do not discuss numerical issues that will be elaborated on in a future publication. Before ending this section, one last note is on the actual implementation of the algorithm to achieve self-consistency. The algorithm outlined above is described just for the conceptual clarity of the approach, but there are more efficient ways to reach self-consistency. Namely, we are using Broyden’s method [13,14] to solve this non-linear set of equations. The equations to solve are basically the free-energy gradients with respect to $\left(u_{0},\eta ,K,C\right)$ being equal to zero. Broyden’s method precisely solves a non-linear system of the form $\vec{G}\left(\vec{X}\right)=0$, with the variables $\left(u_{0},\eta ,K,C\right)$ being stored in an array $\vec{X}$, and the array $\vec{G}$ containing the free energy gradients with respect to $\vec{X}$. After every iteration of Broyden, where all $\left(u_{0},\eta ,K,C\right)$ are updated at the same time, the displacement autocorrelation matrix $\langle y_{i}y_{j}\rangle$ is calculated from the eigenvalues and eigenvectors of $K_{ij}$ using Equation (Equation 7) and is used to calculate the gradients $\vec{G}$, which are fed back into the Broyden algorithm for the next iteration. Non-positivity of the dynamical matrix can be dealt with by taking atomic displacements along the softest modes and shear deformations along $-\frac{\partial F_{trial}}{\partial \eta^{\alpha \beta }}$ to lower the trial free energy until a minimum is reached along that direction. The flowchart of this approach is displayed in Figure 1.

Using this scheme, not only is it possible to obtain the phonon spectrum at finite temperatures, but one may also obtain the equilibrium shape of the unit cell and atomic positions defined by $\eta ,u^{0}$. Thus, this method can predict solid–solid phase transitions as a function of temperature (and also imposed pressure). Two simple implementations of this method have been included in our previous paper [15,16]. One is a single particle in an anharmonic quartic potential in which the change of the equilibrium position and the vibrational frequency versus temperature is illustrated. The other is the treatment of the phase change in a one-dimensional chain with up to sixth-order onsite potential energy and up to quartic nearest-neighbor pair interactions. At low temperatures, the equilibrium position is away from the lattice site, while above the critical temperature, it has a discontinuous jump to the lattice site.

### 2.2. Previous Works/Implementations by Other Groups

We need to mention other similar works in which self-consistent phonon approximation has been implemented from a set of first-principles DFT calculations on forces as a function of displacements, and we point out the small variations compared to the present work. To the best of our knowledge, these include, in chronological order:The SCAILD (Self Consistent Ab Initio Lattice Dynamics) method by Souvatzis et al. in 2008 [17,18], in which the effective dynamical matrix was extracted from the Fourier transform of the forces and displacements in the reciprocal space.TDEP (Temperature-Dependent Effective Potential) method by Hellman et al. in 2011 [19,20,21], in which both harmonic and cubic force constants are extracted from a molecular dynamics simulation performed at several temperatures in a supercell.SSCHA (Stochastic Self-Consistent Harmonic Approximation) method by Errea et al. in 2014 [22,23,24,25], where a reweighting of new configurations is done, and importance sampling is performed to calculate new thermal averages.SCP implemented in the code ALAMODE–ANPHON by Tadano et al. in 2015 [26], where anharmonicity up to the fourth-order is extracted, similar to the present approach, and the self-consistent harmonic approximation is implemented.SCP is also implemented in the HiPHive package developed by Eriksson et al. [27].SCHA (Self-Consistent Harmonic Approximation) method by Ravichandran and Broido in 2018 [28] also calculates the effective phonon dispersion and lifetimes at finite temperatures.Finally, QSCAILD (Quantum Self Consistent Ab Initio Lattice Dynamics) by van Roekeghem et al. in 2021 [29,30,31] is another recent implementation of the method.

While they all enforce self-consistency between the atomic displacements and the effective harmonic force constants, they differ in the way the phase space is sampled and force constants are calculated. The numerical challenges we outlined, especially near a phase transition, still exist in all these methods. In addition, many of these methods, with the exception of SSCHA, do not implement thermal expansion, meaning the effective temperature-dependent strain tensor η and residual atomic relaxations $u_{i}^{0}$ are often not calculated self-consistently unless supercell DFT energy minimization is performed between updates.

### 2.3. Summary

To conclude, we have developed a formalism based on the self-consistent phonon theory in which thermodynamic properties of a crystalline material are computed from the knowledge of the higher-order anharmonic force constants (up to quartic has been illustrated in this paper). It is therefore possible to predict the phonon dispersion, thermal expansion and equilibrium shape of the unit cell and the atoms therein (including phase change) as a function of temperature. The novelty of the approach is that all thermodynamic calculations are done from knowledge of only the force constants of the high-symmetry phase. The limits of the validity of this approach will be tested in a future publication. It is worth mentioning that the implementation as outlined in this paper is much faster than DFT calculations in a supercell, as all averages are Gaussian and can be calculated analytically. The disadvantage is the accuracy of the Taylor expansion force field, which is truncated to fourth-order and up to some nearest-neighbors interactions. In this case, either the order of expansion has to be increased to sixth or higher, or new force constants ϕ,ψ,χ need to be calculated for the new phase in order to calculate its thermodynamic properties (and not its phase change).

## 3. Non-Equilibrium Heat Transport in Anharmonic Systems

The previous section dealt with equilibrium properties at finite temperatures. In this section, however, we are interested in heat current flow under an applied temperature difference across a device. Thus, this will be a non-equilibrium situation, but the formalism will have some similarities to the equilibrium case in terms of thermal expansion and effective force constants. Thermal averages need to be computed, but they will be non-equilibrium thermal averages, for which we use a different approach based on the equation-of-motion method, the fluctuation–dissipation theorem and the Donsker–Furutsu–Novikov theorem [32,33,34], which we describe in detail in the following.

Transport theories of non-interacting quantum systems based on the Keldysh formalism, which treats non-equilibrium flow of charge or heat, have been developed in the past for a one-dimensional (1D) geometry. In this review, we propose a simplified version in the classical or high-temperature limit using a lighter formalism based on the equation-of-motion method.

To the best of our knowledge, the first quantum development of an electron transport theory applied to nanostructures across a junction was done in 1971 by Caroli et al. [35] where a Green’s function formalism was used to describe transmission of electrons in a 1D system. Following the seminal work of Caroli et al, many other groups worked on similar formalisms and proved a formula for the transmission through the system, which is now widely used for both non-interacting electrons and phonons. The equilibrium version of it, namely $T=\mathrm{Tr}[G\Gamma_{L}G^{†}\Gamma_{R}]$, where *T*, *G* and Γ are, respectively, the transmission, the retarded Green’s function and the escape rates to the leads, was established by Meir and Wingreen [36], and in a similar form, by Pastawski [37] in 1991. This formula holds for a non-interacting (in the case of electrons) or harmonic (in the case of phonons) system *near* equilibrium, meaning the chemical potential or temperature gradients are also infinitesimally small. These assumptions might not always be realistic, especially in small (mesoscopic) systems subject to temperature differences over fractions of a micrometer, and a formulation for non-equilibrium situations and interacting systems is preferable for the sake of testing the domain of the validity of the equilibrium formulas and for more accurate description in the case of large interactions and large driving fields.

In this review, we show a semi-classical atomistic derivation of the transport theory of heat carriers in a solid across a junction connected to thermal reservoirs based on Green’s function formalism. The basic geometry of our problem is multi-probe where the system in which scatterings occur is connected to multiple reservoirs, contacts or heat sinks that impose their temperature and cause flow of heat carriers (see Figure 2). This model is used for mesoscopic systems where the carrier mean free path could be on the order of the system length, implying that Ohm’s law of addition of resistances in series does not necessarily hold, and coherence can play an important role. It is also worth noting that since we are in the non-equilibrium regime in general, temperature or chemical potential are not necessarily well-defined concepts, and thus will be avoided. The geometry of the reservoirs is fundamentally one-dimensional (1D), and if there is translational symmetry perpendicular to the current flow, one can use Bloch’s theorem to decouple the 3D system into many non-interacting 1D systems, each labeled by a quantum number, which is the transverse momentum. So in what follows, we assume such decoupling has been done, and that we will be dealing with strictly 1D semi-infinite leads, although the central device is arbitrary in shape and structure and may be connected to multiple 1D probes. In this paper, we are interested in transport of anharmonic phonons, or, more generally, vibrational modes, in mesoscopic systems under large temperature differences. For simplicity, we use a *classical* description. A generalization to the quantum case will be inferred at the end. So in our classical treatment, the frequency ω is just the frequency of vibrational modes and not the energy of phonons. This classical formalism avoids fancier mathematics involving commutation relations and concepts such as time-ordered or contour-ordered Green’s functions. It will only involve "retarded" or causal Green’s functions, which helps us solve a differential equation in the frequency domain. Typically considered geometries will be identical to a non-equilibrium molecular dynamics (NEMD) setup, where the two ends of the system are attached to two thermostats at different temperatures, and one is interested in measuring the interfacial thermal conductance (see Figure 2).

The non-equilibrium anharmonic phonon problem has been addressed in the past by Mingo [38] and separately by Wang [39] in 2006. They used the many-body perturbation approach of non-equilibrium quantum systems based on the Keldysh formalism, (also called the Non-Equilibrium Green’s Function or NEGF method) and derived a lowest-order approximation for the transmission function. Dai and Tian [40] recently implemented this rigorous formulation to calculate the effect of cubic anharmonicity on the phonon transmission function through an ideal interface by applying it to Si/Ge and Al/Al with two different masses. A similar model that explicitly incorporates transverse momentum dependence was also developed recently by Guo et. al. [41] based on previous work by Luisier [42]. Polanco has a recent review of these methods based on NEGF [43]. These NEGF-based models, although fully quantum mechanical, do not include anharmonicity beyond cubic order nor any thermal expansion effects. In this work, we intend to go beyond cubic and include thermal expansion effects self-consistently.

Other calculations of transmission in the non-equilibrium regime [44] based on the Green’s function method have been based on self-consistent reservoirs (also called the Buttiker probe method,) which was first proposed by Bolsterli et al. in 1970 [45]. In this method, to obtain the local temperature, an atom is connected with a *very weak* coupling to a fictitious probe at a given temperature with which it enters equilibrium. The coupling to the probe is weak so that it does not disturb the heat flow within the system. The probe temperature is changed until the net heat current from the system to the added fictitious probe is zero. Therefore, because there is no flow, the probe is in equilibrium with that atom, and the probe temperature defines the temperature of the atom. Note that the system could be out of equilibrium, so the temperature is not really well-defined on a given atom. This shortcoming is also present in NEMD simulations where the assigned “temperature” is just the average kinetic energy of the atom although there is no evidence of local thermal equilibrium. So even though non-equilibrium effects are included through this Buttiker probe method, it does not include anharmonicity. We should mention at this point that there is numerical evidence of absence of equipartition in the vibrational modes near the interface [46,47], implying that a definition of local temperature is not really justified near an interface where the drop could be large and therefore ∇T is not a small perturbation.

In another work, to include anharmonic corrections in the transmission, obtained from MD trajectories, Saaskilahti et al. [48] extended the harmonic formulation of the transmission function by Chalopin et al. [49,50]. In the actual calculation, arguing that the anharmonic part of the current is usually small, they only used its harmonic formula but with velocities and positions coming from the full anharmonic atomic trajectories. The advantage of this approach over NEMD is that the heat current and thermal conductance can be decomposed in the frequency domain. Approaches based on MD trajectories, while including full anharmonicity, suffer from noise and would require a large number of simulations in order to perform proper ensemble averaging, whereas many-body approaches might be inaccurate if a perturbative expansion in powers of anharmonicity is used; otherwise, they do not suffer from noise and treat ensemble averaging analytically. In this review, we show how some of these limitations can be overcome by adopting a non-perturbative self-consistent many-body approach by fully including in the current the effect of anharmonic terms introduced in the Hamiltonian. Furthermore, using a classical method to derive an expression for the heat current, we argue that to leading order, it is necessary to include quartic terms in the Hamiltonian to ensure stability and properly describe both the thermal expansion and the dominant temperature dependence effect on the heat current.

### 3.1. Equations of Motion and Langevin Thermostats

We start by defining our model and the assumptions. A multiprobe geometry is assumed. Figure 2 shows a two-probe example, in which a central region, also called the “device”, is connected to two semi-infinite one-dimensional (1D) leads that play the role of thermostats imposing a temperature at the boundaries of the system. For a 3D geometry where periodic boundary conditions are used in the transverse direction, i.e., perpendicular to the heat flow direction, as shown in the figure, a Fourier transformation will reduce it to $m^{2}$ one-dimensional problems, where *m* is the number of transverse kpoints. We are not concerned with temperature drops in the thermostats, which are assumed to be harmonic heat sinks and follow Langevin dynamics.

The anharmonic Hamiltonian of the device (D) and its coupling to the lead or probe α are:(23)$$\begin{array}{cc} H_{D} & =\sum_{i\in D}\frac{p_{i}^{2}}{2m_{i}}+\sum_{ij\in D}\;\frac{1}{2!}\phi_{ij}\;u_{i}\;u_{j}+\sum_{ijk\in D}\;\frac{1}{3!}\psi_{ijk}\;u_{i}\;u_{j}\;u_{k}+... \end{array}$$
(24)$$\begin{array}{cc} H_{\alpha ,D} & =\sum_{i\in D}\sum_{l\in \alpha }\;W_{\alpha ,il}\;u_{i}\;u_{l} \end{array}$$
The dynamical variable $u_{i}\left(t\right)$ refers to the displacement of atom *i* about the zero-temperature equilibrium position (the force on the atom is zero for u=0). The leads labeled by α are semi-infinite chains following Langevin dynamics. After the standard change of variables to $x_{i}\left(t\right)=\sqrt{m_{i}}u_{i}\left(t\right)$, and with
$$\Phi_{ij}=\frac{\phi_{ij}}{\sqrt{m_{i}m_{j}}}=\frac{1}{\sqrt{m_{i}m_{j}}}\frac{\partial^{2}H_{D}}{\partial u_{i}\partial u_{j}}\left(u=0\right)$$

$V_{\alpha ,il}=W_{\alpha ,il}/\sqrt{m_{i}m_{l\alpha }}$ and $\Phi_{\alpha ,ll^{\prime }}=\phi_{\alpha ,ll^{\prime }}/m_{l\alpha }$, we arrive at the following equation of motion for atoms in the central region:(25)$$\frac{d^{2}x}{dt^{2}}=-\Phi \;x-\sum_{\alpha }V_{\alpha }\;x_{\alpha }+a$$

Note we use capitalized Greek letters (Φ,Ψ,…) for mass-rescaled force constants and lower-case Greek letters (ϕ,ψ,…) for the bare potential energy derivatives. The letter α refers to the leads, and the dynamical variable $x=(x_{1},\ldots ,x_{N})$ can be thought of as an array containing the displacements of all the atoms in the central region (also called the device), Φ as the force constant matrix between such atoms, and $V_{\alpha }$ as the force constant matrix connecting atoms of the lead α to atoms in the device. Finally, $a=-1/2\Psi x^{2}+\ldots$ is the anharmonic part of the force, which, for now, we keep as *a* for brevity. The dynamics of atoms in the lead are of *Langevin* type, where a set of identical coupled harmonic oscillators are subject to damping $\gamma_{\alpha }$ and noise $\zeta_{\alpha }$ as follows:(26)$$\frac{d^{2}x_{\alpha }}{dt^{2}}=-\Phi_{\alpha }x_{\alpha }-V_{\alpha }^{T}x-\gamma_{\alpha }\frac{dx_{\alpha }}{dt}+\zeta_{\alpha }$$
where the superscript *T* stands for transpose. The force constants $\Phi_{\alpha }$ can be thought of as effective FCs at the temperature of interest, so that we do not need to introduce anharmonicity in the leads, which merely play the role of absorbing phonons from the device and reinjecting thermalized phonons into the device. To insure each thermostat has temperature $T_{\alpha }$, the damping coefficient γ and the random forces ζ must satisfy the following fluctuation–dissipation relation in Equation (Equation 27):$$\langle \zeta_{i,\alpha }\left(t\right)\rangle =0$$
(27)$$\langle \zeta_{i,\alpha }\left(t\right)\zeta_{j,\alpha^{\prime }}\left(t^{\prime }\right)\rangle =2\gamma_{\alpha }k_{B}T_{\alpha }\;\delta \left(t-t^{\prime }\right)\delta_{\alpha ,\alpha^{\prime }}\delta_{ij}$$

We proceed by eliminating the lead variables $x_{\alpha }$ in Equation (Equation 25) using the Green’s function method. To this end, we start by taking the Fourier transform of the above two equations according to:(28)$$X\left(\omega \right)=\int dt\;x\left(t\right)\;e^{i\omega t};\;\;x\left(t\right)=\int \;\frac{d\omega }{2\pi }\;X\left(\omega \right)\;e^{-i\omega t}$$

The equations of motion satisfied by the Fourier transform of the displacements become:(29)$$\begin{array}{cc} -\omega^{2}X & =-\Phi X-\Sigma_{\alpha }V_{\alpha }\;X_{\alpha }+A \end{array}$$
(30)$$\begin{array}{cc} -\omega^{2}X_{\alpha } & =-\Phi_{\alpha }X_{\alpha }-V_{\alpha }^{T}X+i\omega \gamma_{\alpha }X_{\alpha }+\zeta_{\alpha }\left(\omega \right) \end{array}$$

Note that the frequency-domain variables are represented with capitalized letters. Now that the differential equations are transformed to algebraic ones, one can easily proceed to eliminate the lead degrees of freedom in the main equation of motion by using the Green’s functions. Let $g_{\alpha }\left(\omega \right)$ be the retarded (causal) Green’s function associated with the lead α. The positivity of the damping factor $\gamma_{\alpha }$ insures causality. The solution to Equation (Equation 30) *after the transients have decayed to zero* can be written as:(31)$$X_{\alpha }=g_{\alpha }\left(\omega \right)\;\left(\zeta_{\alpha }-V_{\alpha }^{T}X\right)$$
where
(32)$$g_{\alpha }^{-1}=\left[-\omega^{2}-i\omega \gamma_{\alpha }+\Phi_{\alpha }\right]$$

In Equation (Equation 29), we need $-V_{\alpha }X_{\alpha }=\eta_{\alpha }+\sigma_{\alpha }X$, which we obtain from Equation (Equation 31). Here, $\eta_{\alpha }\left(\omega \right)=-V_{\alpha }g_{\alpha }\left(\omega \right)\zeta_{\alpha }\left(\omega \right)$, and $\sigma_{\alpha }=V_{\alpha }\;g_{\alpha }V_{\alpha }^{T}$. Likewise, defining the retarded Green’s function of the central region as $G^{-1}=\left[-\omega^{2}+\Phi -\sum_{\alpha }\sigma_{\alpha }\left(\omega \right)\right]$, we can write the solution to the central region as:(33)$$X\left(\omega \right)=G\left(\omega \right)\;\left[\sum_{\alpha }\eta_{\alpha }\left(\omega \right)\;+A\left(\omega \right)\right]$$

The function $\sigma_{\alpha }\left(\omega \right)=V_{\alpha }\;g_{\alpha }\left(\omega \right)V_{\alpha }^{T}$ is traditionally called the self-energy of lead α, and it seems like an added correction to the harmonic FC’s Φ. Its effect is to shift the vibrational frequencies of the device and give them a a finite lifetime, as these modes can leak into the leads. Omitting the transient contribution of the initial conditions, this is the solution to the equations of motion, which depend on the stochastic functions $\eta_{\alpha }=-V_{\alpha }g_{\alpha }\zeta_{\alpha }$.

To summarize, as is also illustrated in Figure 2, the thermostats have been eliminated and replaced by Langevin forces $\eta_{\alpha }$ applied directly on the device atoms connected to a thermostat, and the harmonic FCs have been renormalized to become $\Phi -\sum_{\alpha }\sigma_{\alpha }$.

### 3.2. Entropy Generation Rate

An important quantity of interest is the entropy generation rate in the device, which in steady state can be expressed in terms of heat fluxes $j_{\alpha }$ as: $\dot{S}=-\sum_{\alpha }\langle j_{\alpha }\rangle /T_{\alpha }$. This quantity results from the interaction between the device degrees of freedom and the random noise due to Langevin thermostats. It involves the thermostat’s temperature and the incoming currents, which will be discussed next.

### 3.3. Heat Current

The heat flow from leads to the device is microscopically defined as the average rate lead atoms do work on the device atoms. The work rate is the product of the velocity degrees of freedom of the device multiplied by the force from the lead acting on them. With this link being harmonic, the heat current expression is simply: $j_{\alpha }\left(t\right)=\langle \mathrm{Tr}\left[\dot{x}\;{\left(-V_{\alpha }x_{\alpha }\right)}^{T}\right]\rangle$, where the trace is taken over the device degrees of freedom, and averaging is over the thermostat degrees of freedom. Since the current depends on the stochastic functions ζ, we need to take the ensemble average over all realizations of the random forces subject to the fluctuation–dissipation theorem [51]. To get the steady-state current, we also take the time average. In terms of its Fourier transform, the time-integrated current is $\langle J_{\alpha }\left(\Omega =0\right)\rangle =\int_{-\infty }^{\infty }\langle j_{\alpha }\left(t\right)\rangle dt$. The current may be calculated using Equations (Equation 31) and (Equation 33), and the result can be written as:(34)$$\langle J_{\alpha }\left(\Omega \right)\rangle =\int \frac{d\omega }{2\pi }\;\omega \;\mathrm{Tr}\;\left[ℑ\langle X\left(\omega \right)\eta_{\alpha }^{†}\left(\omega -\Omega \right)\rangle -\langle X\left(\omega \right)X^{†}\left(\omega -\Omega \right)\rangle \;ℑ\sigma_{\alpha }\left(\omega -\Omega \right)\right]$$

The DC response is found by taking the Ω→0 limit and dividing by the time-integration parameter τ→∞. As shown later, the results for $\langle j_{\alpha }\rangle$ do not depend on τ. Note that because we are interested in the DC response, only diagonal terms of correlations (Ω=0) are needed here. Thus, calculation of the heat current is reduced to calculation of the two correlation functions and the so-called lead self-energy $\sigma_{\alpha }\left(\omega \right)$, followed by a frequency integration. Note that this current from lead α is the sum of two terms: the first one, proportional to $Z_{\alpha }=\langle X\left(\omega \right)\;\eta_{\alpha }^{†}\left(\omega \right)\rangle$, is the work of the stochastic forces on the device; while the second one, proportional to a displacement autocorrelation $\langle X\left(\omega \right)\;X^{†}\left(\omega \right)\rangle$, is the work of the lead dampers trying to reabsorb some of the excess energy injected from the device in order to reestablish thermal equilibrium in the lead.

The average denoted by 〈〉 is an average of the stochastic forces in the thermostats, which have white noise characteristics. With the leads being at different temperatures, the average becomes a *non-equilibrium* average and can only be calculated using the equation of motion (Equation 33) and the statistical properties of the Langevin thermostats following the fluctuation–dissipation theorem. For anharmonic interactions involving higher powers of displacements in *A*, the calculation of current leads to a hierarchy of equations, each containing higher powers of displacements, and has, so far, been computed using different approximations [38,39,52].

One should note that at equal lead temperatures, one is in equilibrium and no net current enters the device: $\langle j_{\alpha }\rangle =0$. If the lead temperatures are different and one is in the non-equilibrium regime, in the absence of heat generation in the device, we should still have $\Sigma_{\alpha }\langle j_{\alpha }\rangle =0$. This is also known as *Kirchhoff’s law* of current conservation in the context of electrical circuits.

### 3.4. Thermal Expansion and Renormalization of the Force Constants

At finite temperatures, equilibrium or not, the average position of the atoms is shifted from its zero-temperature reference value because of anharmonicity. Treatment of the thermal expansion is similar to that of the equilibrium case, where solving Equation (Equation 16) for given $\langle y_{i}y_{j}\rangle$ leads to the shifts *s*, with the exception that the thermal averages are now non-equilibrium averages. So first, we proceed to calculate the thermal expansion and the resulting renormalization of the force constants.

The set of equations expressing that the force on each atom is, on average (over time and over random Langevin forces), zero $\langle \frac{\partial H_{D}}{\partial x_{i}}\rangle =0$, leads to the average atomic positions $\langle x_{i}\rangle =s_{i}$ satisfying the following cubic equation in the parameters *s*:(35)$$\begin{array}{c} \sum_{j}\left(\phi_{ij}+\frac{1}{2}\sum_{kl}\chi_{ijkl}\langle y_{k}y_{l}\rangle \right)\;s_{j}+\frac{1}{2}\sum_{jk}\psi_{ijk}\;s_{j}s_{k}+\frac{1}{6}\sum_{kjl}\chi_{ijkl}s_{j}s_{k}s_{l}=-\frac{1}{2}\sum_{jk}\psi_{ijk}\;\langle y_{j}y_{k}\rangle \end{array}$$

Due to these position shifts, as well as thermal fluctuations of atomic positions, the force constants effectively change, similar to the SCP case. One can show that the effective harmonic FCs will still be given by Equation (Equation 15), again, with the exception that the thermal average is a non-equilibrium one. In this case, we change displacement variables from *x* to *y* such that y(t)=x(t)−〈x〉=x(t)−s, which has zero average by construction. The resulting equations of motion satisfied by *y* can be shown to be:(36)$$-\omega^{2}Y=-\left(K-\sum_{\alpha }\sigma_{\alpha }\right)Y+A+\sum_{\alpha }\eta_{\alpha }$$
where the renormalized anharmonic force is
$$A\left(\omega \right)=-\frac{1}{2}\bar{\Psi }\sum_{\Omega }Y\left(\omega -\Omega \right)Y\left(\Omega \right)-\frac{1}{6}\chi \sum_{\Omega_{1}\Omega_{2}}Y\left(\omega -\Omega_{1}-\Omega_{2}\right)\times$$
$$\left[Y\left(\Omega_{1}\right)Y\left(\Omega_{2}\right)-3\langle Y\left(\Omega_{1}\right)Y\left(-\Omega_{1}\right)\rangle \delta_{\Omega_{1},-\Omega_{2}}\right]$$
and the renormalized cubic force constant is $\bar{\Psi }=\Psi +\chi s$. (The quartic term is affected if there are fifth or higher derivatives of the potential. In this approximation where we limit ourselves to the quartic term in the expansion of the potential energy, the latter is not renormalized.) The renormalized Green’s function corresponding to this equation of motion is therefore defined as
(37)$$\begin{array}{c} G\left(\omega \right)={\left[-\omega^{2}+K-\sum_{\alpha }\sigma_{\alpha }\right]}^{-1} \end{array}$$
leading to the general solution:(38)$$\begin{array}{c} Y=G(\sum_{\alpha }\eta_{\alpha }+A) \end{array}$$

One could call this the non-equilibrium mean-field, or the *non-equilibrium self-consistent phonon* (NESCP) approximation. In our original paper on this topic [53], we explain that this is the “best” mean-field approximation, as it includes the mean-field corrections $-\langle \frac{\partial A}{\partial Y}\rangle =\frac{1}{2}\chi \langle YY\rangle$ in the harmonic force constants and leads to leading-order cancellation of anharmonic terms in the expansion of the correlation functions $\langle Y\eta^{†}\rangle$ and $\langle YY^{†}\rangle$ in powers of $\frac{\partial A}{\partial Y}$ and A, respectively.

### 3.5. Correlation Functions $\langle Y\eta^{†}\rangle$ and $\langle YY^{†}\rangle$

Now, we proceed to the calculation of the two needed correlation functions $\langle Y\eta^{†}\rangle$ and $\langle YY^{†}\rangle$ in order to obtain the heat current $\langle j_{\alpha }\rangle$, given as below.
(39)$$\begin{array}{c} \langle j_{\alpha }\rangle =\lim_{\overset{\tau \to \infty }{\Omega \to 0}}\frac{\langle J_{\alpha }\left(\Omega \right)\rangle }{\tau }=\int \frac{d\omega }{2\pi }\;\omega \;\left[\frac{ℑ\langle Y\left(\omega \right)\eta_{\alpha }^{†}\left(\omega \right)\rangle }{\tau }+\frac{\langle Y\left(\omega \right)Y^{†}\left(\omega \right)\rangle }{\tau }\;ℑ\sigma_{\alpha }^{†}\left(\omega \right)\right] \end{array}$$

#### 3.5.1. Displacement—Noise Correlations $\langle Y\eta^{†}\rangle$ within NESCP

Let $Z_{\alpha }\left(\omega \right)=\langle Y\left(\omega \right)\eta_{\alpha }^{†}\left(\omega \right)\rangle /\tau$. The product $\omega ℑ(Z_{\alpha })$ is the power exerted by the random force η on the device and can therefore be interpreted as the *heat injected per unit time and unit frequency (mode) from lead α into the device*. We can calculate this term using the solution to the equation of motion, Equation (Equation 38).

In the NESCP approximation, A is neglected, and we have $Z_{\alpha }^{NESCP}=G\langle \eta_{\alpha }\eta_{\alpha }^{†}\rangle /\tau$. The noise autocorrelation is obtained from the fluctuation–dissipation theorem, Equation (Equation 27), and it leads to (see Section B.2, Equation (Equation 60)):(40)$$\begin{array}{c} \frac{\langle \eta_{\alpha }\left(\omega \right)\eta_{\alpha }^{†}\left(\omega \right)\rangle }{\tau }=\Gamma_{\alpha }\left(\omega \right)\frac{k_{B}T_{\alpha }}{\omega }=\Gamma_{\alpha }\left(\omega \right)f_{\alpha } \end{array}$$
where, for brevity, we replaced the “occupation factor” $k_{B}T_{\alpha }/\omega$ with $f_{\alpha }$, *(With a factor of ℏ in the denominator $f_{\alpha }$ would be a real Bose-Einstein occupation factor taken to the classical limit)* and τ represents the integration time, which goes to infinity and is canceled in the final expressions. For white noise, one can show that the result does not depend on the thermostat damping parameter γ, and thus, we adopt this type of noise for the thermostats. Finally, within NESCP, the displacement noise correlation function is given by:(41)$$Z_{\alpha }^{NESCP}\left(\omega \right)=G\left(\omega \right)\Gamma_{\alpha }\left(\omega \right)f_{\alpha }$$

#### 3.5.2. Beyond NESCP

To go one step further beyond SCP and include the effect of anharmonicity in $Z_{\alpha }$, we use the Donsker–Furutsu–Novikov (DFN) identity [54] (for a proof, see Appendix C), which states that for any functional of the white noise f[η], we have
(42)$$\langle f\left[\eta \right]\;\eta_{\alpha }^{†}\left(\omega \right)\rangle =\langle \frac{\delta f[\eta ]}{\delta \eta_{\alpha }}\rangle \langle \eta_{\alpha }\;\eta_{\alpha }^{†}\rangle$$

It has the advantage of lowering the powers of η in *f*. Using this theorem, we have: $\langle Y\eta_{\alpha }^{†}\rangle =\langle \frac{\partial Y}{\partial \eta_{\alpha }}\rangle \langle \eta_{\alpha }\;\eta_{\alpha }^{†}\rangle$. From the equation of motion for *Y* and the chain rule, we find
$$\begin{array}{c} \frac{\partial Y}{\partial \eta_{\alpha }}={\left(1-G\frac{\partial A}{\partial Y}\right)}^{-1}G=G+G\frac{\partial A}{\partial Y}G+G\frac{\partial A}{\partial Y}G\frac{\partial A}{\partial Y}G+... \end{array}$$
so that finally,
(43)$$\begin{array}{ccc} Z_{\alpha } & = & \langle Y\eta_{\alpha }^{†}\rangle =\langle \frac{\partial Y}{\partial \eta_{\alpha }}\rangle \langle \eta_{\alpha }\;\eta_{\alpha }^{†}\rangle =\langle {\left(1-G\frac{\partial A}{\partial Y}\right)}^{-1}\rangle \;Z_{\alpha }^{NESCP}\\ & = & \left(1+G\langle \frac{\partial A}{\partial Y}G\frac{\partial A}{\partial Y}\rangle +G\langle \frac{\partial A}{\partial Y}G\frac{\partial A}{\partial Y}G\frac{\partial A}{\partial Y}\rangle +...\right)\;Z_{\alpha }^{NESCP} \end{array}$$
This is an *exact* relation satisfied by $Z_{\alpha }$. Note that, by construction, the first term involving G〈∂A/∂Y〉G is identically zero because 〈ΨY〉=0 and 〈χYY〉/2 is already included in *K*, which appears in the denominator of G. More details about the proof can be found in [53]. In the language of many-body theory, this is the expanded form of Dyson’s equation involving the thermal average of powers of the anharmonic force derivatives. This result, even though exact, cannot be calculated because the average of the inverse of powers of *Y* cannot be calculated exactly. Instead, in the spirit of perturbation theory, one may add the power-series term-by-term provided the sum is convergent. Another non-perturbative approach, which we adopt here, is to sum a series of terms up to infinite order.
(44)$$Z_{\alpha }\approx {\left(1-G\langle \frac{\partial A}{\partial Y}G\frac{\partial A}{\partial Y}\rangle \right)}^{-1}\;Z_{\alpha }^{NESCP}$$
with
(45)$$A=A-Y\langle \frac{\partial A}{\partial Y}\rangle =-\frac{1}{2}\bar{\Psi }YY-\frac{1}{6}\chi \left(YYY-3Y\langle YY\rangle \right)$$
and $\frac{\partial A}{\partial Y}=\frac{\partial A}{\partial Y}-\langle \frac{\partial A}{\partial Y}\rangle =-\bar{\Psi }Y-\frac{1}{2}\chi \left(YY-\langle YY\rangle \right)$, which has a zero average by construction. We can note that the major part of the quartic anharmonicity has been removed, and the expansion is in powers of ∂A/∂Y, which is centered at zero and therefore has the smallest higher moments. When raised to second power in the above formula for $Z_{\alpha }$, cubic and quartic terms become decoupled if we neglect averages of terms of odd power in *Y*, which are expected to be small if non-zero. At low temperatures or weak anharmonicity, the dominant contribution to the second-order terms comes from cubic terms and can be written as:(46)$$Z_{\alpha }\approx {\left(1-G\bar{\Psi }G\bar{\Psi }\langle YY\rangle \right)}^{-1}\;Z_{\alpha }^{NESCP}$$

#### 3.5.3. Displacement Autocorrelations $\langle YY^{†}\rangle$

Finally, the last correlation function needed is $C\left(\omega \right)=\langle Y\left(\omega \right)Y^{†}\left(\omega \right)\rangle /\tau$. Note the autocorrelation matrix *C* is Hermitian. As before, within the SCP, where the effective force constant *K* appears in the denominator of G, one arrives at a simple expression for *C*:(47)$$C^{NESCP}\left(\omega \right)=G\;\Sigma_{\alpha }\langle \eta_{\alpha }\eta_{\alpha }^{†}\rangle \;G^{†}/\tau =\Sigma_{\alpha }\;\left(G\Gamma_{\alpha }G^{†}\right)\;f_{\alpha }$$

Let us first give a physical interpretation of this function as it is related to the heat current. As shown in Equation (A6) in the Section B.2, if $(\Sigma_{\alpha }\Gamma_{\alpha })=\Gamma$ is twice the imaginary part of the denominator of the Green’s function G, then $G\Gamma G^{†}=2ℑ\left(G\right)$, so that at equilibrium, where all leads are at the same temperature, $C^{NESCP}=2ℑGf_{\mathrm{eq}}$. On the other hand, ℑG is related to the vibrational density of states (DOS) through:$$-\frac{2\omega }{\pi }\;\mathrm{Tr}\;ℑG\left(\omega \right)=\sum_{\lambda }2\omega \;\delta \left(\omega^{2}-\omega_{\lambda }^{2}\right)=\sum_{\lambda }\delta \left(\omega -\omega_{\lambda }\right)=DOS\left(\omega \right)$$

Therefore, if all lead temperatures are equal, the function
$$DOS\left(\omega \right)\;f_{\mathrm{eq}}\left(\omega \right)/\hbar =-\omega \mathrm{Tr}\;C^{NESCP}\left(\omega \right)/\pi \hbar$$
can be interpreted as the equilibrium number of vibrational excitations of frequency ω present in the device due to its contact with the leads (Since $f_{\alpha }=kT_{\alpha }/\omega$, then $f_{\alpha }/\hbar =kT_{\alpha }/\hbar \omega$ is the classical (high-temperature) limit of the equilibrium Bose–Einstein occupation factor, and multiplied by the DOS, it can be interpreted as the number of vibrational excitations of frequency ω.). On the other hand, as the imaginary part of the device’s Green’s function $\Gamma_{\alpha }/2\omega$ can be understood as the rate of absorption of these vibrational excitations by the leads, we can say that $-\mathrm{Tr}C^{NESCP}\Gamma_{\alpha }/2\pi \hbar$ is the rate of flow of phonons of frequency ω going from the device into the lead α. The heat current can be obtained by multiplying this particle current by the energy ℏω of each carrier, leading to $\omega \mathrm{Tr}C^{NESCP}\Gamma_{\alpha }/2\pi$ being the heat flowing from device to lead α. This is indeed the second term in the expression of the heat current in Equation (Equation 34), the first term being the heat flow from the lead into the device.

#### 3.5.4. Beyond NESCP

Now, we can go one step beyond NESCP and, using the solution to the equation of motion (Equation (Equation 38)), express the *exact formula* for the displacement autocorrelation as:(48)$$\begin{array}{cc} & C\left(\omega \right)=\langle Y\left(\omega \right)Y^{†}\left(\omega \right)\rangle =\\ & G\left(\omega \right)\left[\Sigma_{\alpha }\left(\langle \eta_{\alpha }\eta_{\alpha }^{†}\rangle +\langle A\eta_{\alpha }^{†}\rangle +\langle \eta_{\alpha }A^{†}\rangle \right)+\langle AA^{†}\rangle \right]G^{†}\left(\omega \right) \end{array}$$

The first term is the NESCP contribution discussed above. The second and third terms involve terms linear in A that also appeared in the calculation of $Z_{\alpha }$, and finally, the last term involves the second power of the anharmonic force, which includes powers of displacement equal to or higher than four (see Equation (Equation 45)), and as such, cannot be exactly calculated unless the non-equilibrium distribution function of displacements *Y* is known.

As for the displacement autocorrelations, they have the terms $\langle A\eta^{†}\rangle$ and $\langle AA^{†}\rangle$ sandwiched between G and $G^{†}$. The former can be shown to exactly satisfy:(49)$$G\langle A\eta_{\alpha }^{†}\rangle =Z_{\alpha }-Z_{\alpha }^{NESCP}=>G\langle A\eta_{\alpha }^{†}\rangle G^{†}=\left(Z_{\alpha }-Z_{\alpha }^{NESCP}\right)G^{†}$$

The last term, to the lowest-order in cubic anharmonicity, is: $\langle AA^{†}\rangle =\bar{\Psi }\bar{\Psi }\langle YYYY\rangle /4$. Not having the distribution function of *Y*, this expression cannot be calculated; but again, to the leading order in anharmonicity, we use a decoupling approximation to write it as:(50)$$\langle AA^{†}\rangle \approx \frac{1}{4}{\bar{\Psi }}^{2}\left[\sum_{\omega_{1}}C\left(\omega -\omega_{1}\right)C\left(\omega_{1}\right)\times 2\right]$$

This approximation becomes exact in the limit where all leads have the same temperature, so it should be reliable for small temperature differences. It is also good in the limit where the four displacement operators are not the same. Finally, collecting all quadratic terms in $\bar{\Psi }$, we have:(51)$$\begin{array}{cc} & Z_{\alpha }=\left(1+2G{\bar{\Psi }}^{2}\sum_{\omega_{1}}G\left(\omega -\omega_{1}\right)C\left(\omega_{1}\right)\right)Z_{\alpha }^{NESCP}\\ & C=C^{NESCP}+\sum_{\alpha }\left(\delta Z_{\alpha }\;G^{†}+G\;\delta Z_{\alpha }^{†}\right)+GPG^{†}\\ & \delta Z_{\alpha }=Z_{\alpha }-Z_{\alpha }^{NESCP}\\ & P=\frac{1}{2}{\bar{\Psi }}^{2}\sum_{\omega_{1}}C\left(\omega -\omega_{1}\right)C\left(\omega_{1}\right) \end{array}$$

The symbolic diagrams summarizing this approximation are shown in Figure 3.

### 3.6. The Particular Case of Non-Equilibrium Self-Consistent Phonon Approximation: NESCP

To summarize, within the NESCP, the renormalized anharmonic force A is neglected, and we end up with simpler expressions for the correlation functions, similar to the harmonic case:(52)$$\begin{array}{ccc} Z_{\alpha }^{NESCP}\left(\omega \right) & = & \langle Y\eta_{\alpha }^{†}\rangle /\tau =G\Gamma_{\alpha }\frac{k_{B}T_{\alpha }}{\omega }; \end{array}$$(53)$$\begin{array}{ccc} C^{NESCP}\left(\omega \right) & = & \langle YY^{†}\rangle /\tau =\Sigma_{\beta }\left(G\Gamma_{\beta }G^{†}\right)\frac{k_{B}T_{\beta }}{\omega } \end{array}$$

The frequency in the denominator cancels the frequency in the current coming from the time derivative of positions, so the NESCP current becomes:$$\langle j_{\alpha }^{NESCP}\rangle =k_{B}\int \frac{d\omega }{2\pi }\;\mathrm{Tr}\;\left(ℑ\left(G\right)\;\Gamma_{\alpha }T_{\alpha }-\Sigma_{\beta }\left(G\;\Gamma_{\beta }T_{\beta }\;G^{†}\right)\frac{\Gamma_{\alpha }}{2}\;\right)$$

It can be shown that this current satisfies both detailed balance and current conservation [53].

As a result, this expression can be simplified to the familiar form:(54)$$\begin{array}{c} \langle j_{\alpha }^{NESCP}\rangle =\Sigma_{\beta }\;\frac{k_{B}}{2}\left(T_{\alpha }-T_{\beta }\right)\;\int \frac{d\omega }{2\pi }\;\mathrm{Tr}\;\left[\Gamma_{\alpha }\;G\;\Gamma_{\beta }G^{†}\right] \end{array}$$

This equation is linear in temperature differences and clearly satisfies Kirkhoff’s law of current conservation, as it is antisymmetric with respect to swapping of lead indices.

In general, $\mathrm{Tr}\;[\Gamma_{\alpha }\;G\;\Gamma_{\beta }\;G^{†}]$ may be interpreted as the transmission from lead α to lead β. While in a harmonic theory, the transmission is temperature-independent, within NESCP, it actually depends on the leads’ temperatures through G, which contains *K*, which itself contains the temperature-dependent term 〈YY〉 or *C* (see Equation (Equation 15)).

To obtain the heat current within the NESCP model, one needs to self-consistently solve Equations (Equation 15), (Equation 35), (Equation 37) and (Equation 47) defining, respectively, $C^{\mathrm{NESCP}}$, thermal expansion *s*, effective force constant *K* and the effective Green’s function G. Once converged, these quantities can be used in Equation (Equation 54) to calculate the current.

## 4. Conclusions

To summarize, in this review, we presented a formalism to describe thermodynamic and thermal transport properties of solids at high temperatures within a lattice dynamics approach. The first part of this review of our recent work concerned the thermal equilibrium properties of crystalline solids. Starting from calculations performed in a supercell, one can extract the force constants [10] and thus define the anharmonic model Hamiltonian. The variational formulation then allowed definition of the thermal equilibrium state: the free energy needs to be minimized with respect to the parameters defining the equilibrium state, namely the primitive cell translation vectors, atomic positions and effective harmonic force constants. Thermodynamic properties, including the phonon frequencies, can then be obtained from the latter.

In the second part, we developed a self-consistent current-conserving approximation for anharmonic systems out of equilibrium in the high-temperature (classical) regime. Although the set of derived Equations for the current (Equation (Equation 34)), the Equation of motion (Equation (Equation 38)), and for defining the correlation functions (Equations (Equation 43) and (Equation 48)) were formally exact, one has to develop approximations to solve Dyson’s Equation (Equation (Equation 43)) and the Equation defining displacement autocorrelations *C* (Equation (Equation 48)). One reason is because the anharmonic force A is an infinite Taylor expansion and is usually truncated, and another reason is because its derivative appears in the denominator of Equation (Equation 43), which cannot be exactly inverted. In this work, we truncated the Taylor expansion of A up to quartic terms and only included up to second powers of A and ∂A/∂Y in Equations (Equation 43) and (Equation 48), leading to the approximation summarized in Equation (Equation 51).

Furthermore, we showed that thermal expansion needs to be included using both cubic and quartic terms (to avoid any divergence), and it has the effect of renormalizing FCs as *T* is increased. The reference GF to work with, G, has two corrections: one due to thermal expansion implying changes in bond length and strength, and the other due to thermal fluctuations about the average position ($\frac{1}{2}\chi \langle YY\rangle$, similar in spirit to the equilibrium self-consistent phonon theory Equation (Equation 15), except that one is not at thermal equilibrium, and averages depend on the temperature of all attached probes).

Non-equilibrium averages were possible to calculate with the use of the fluctuation–dissipation theorem (Equation (Equation 40)), the equations of motion (Equation (Equation 38)) and the DFN theorem (Equation (Equation 42)). Within the NESCP, where A is dropped from the equation of motion, we end up with a simple description that includes only the average of quartic anharmonicity in the effective force constants. We argue that besides the thermal expansion, this is the leading anharmonic correction to the thermal current.

An alternative approach to investigate non-equilibrium effects at and near interfaces would be to perform a non-equilibrium molecular dynamics simulation (NEMD) of the system attached to thermostats at different temperatures and sample the atomic trajectories in the phase space to find the distribution functions, position averages and force constant averages by fitting the forces to a linear model $F_{i}^{NEMD}\approx F_{i}^{Harmonic}=-K_{ij}y_{j}$ based on the knowledge of the forces on atoms and their positions in each MD snapshot in order to extract the effective (non-equilibrium) harmonic force constants *K*. The remainder can then be defined as the anharmonic force $F_{i}^{NEMD}=-K_{ij}y_{j}+a_{i}\left(y\right)$, and the present results maybe used. This approach, however, has inherent noise in it. If error in the averages is small, one still can use Equation (Equation 54) to compute current if there is no access to the heat current expression in terms of interactions. Another possibility is to use the approximate harmonic formula of the heat current in order to evaluate it [48].

Despite the approximations used in this work, the advantage of this formalism over MD simulations that includes anharmonicity to all orders is that it is analytical and therefore fast and free of simulation noise, although reaching self-consistency can be challenging for some model systems. It would be desirable to compare the results with those of NEMD to validate these approximations for a given system. The accuracy also relies on the force field and strength of higher-order terms: sources of divergence would be in the denominator of Equation (Equation 43) if G〈∂A/∂Y〉≥1, signaling resonances, in which case the Taylor expansion in Equation (Equation 43) may not be appropriate. Applications of this methodology to nanoscale systems and interfaces will appear in future publications.

## Figures and Tables

**Figure 1 entropy-24-01585-f001:**
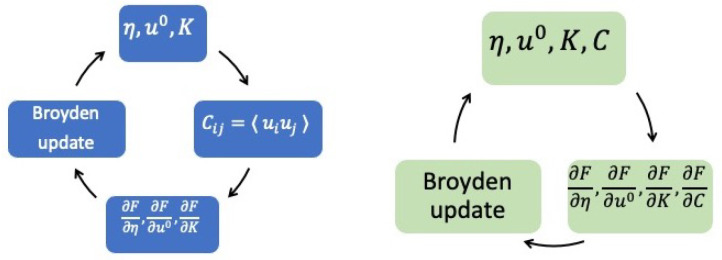
Flowcharts of two possible self-consistent procedures using Broyden’s algorithm depending on whether *K* and *C* are treated as dependent (**left**) or independent parameters (**right**).

**Figure 2 entropy-24-01585-f002:**
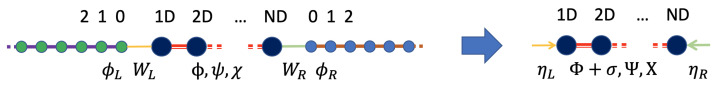
The ND atom anharmonic structure is connected to two semi-infinite reservoirs in this example (called L and R). After elimination of the reservoirs’ degrees of freedom, the infinite system is replaced by an isolated “cluster” subject to thermostating forces $\eta_{L}$ and $\eta_{R}$ applied on the boundary, reflecting the equation of motion in Equation (Equation 33). As a result of this boundary condition, force constants ϕ,ψ,χ are replaced with effective force constants Φ,Ψ,X and a harmonic self energy σ.

**Figure 3 entropy-24-01585-f003:**
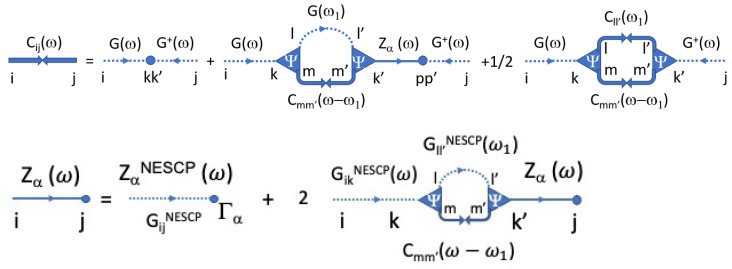
Feynman diagrams associated with the approximations of *C* and $Z_{\alpha }$: dashed lines represent G, the Green’s function within NESCP; thick solid lines with two arrows facing each other represent the displacement autocorrelation *C*; and the solid lines ending with a dot are the noise-displacement correlation $Z_{\alpha }$.

## Data Availability

Not applicable.

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
