# Peer review of "Equilibrium and Non-Equilibrium Lattice Dynamics of Anharmonic Systems"

_entropy, 2022, doi:10.3390/e24111585_

Round 1

Reviewer 1 Report

This is a neat review of the physical models and mathematical description. The work details the development of multiple levels of lattice dynamics theory with increasing level of complexity. The theory of non-equilibrium anharmonic lattice dynamics is of critical importance to broaden our understanding of many problems in diverse context.

Reading this article, I can imagine the hard work and many hours of literature search by the authors, for which I would express my sincere appreciation.

Here is my recommendation. While the review listed a number of development citing precedent literature, however, the merit of the original contribution of this review is not entirely visible through the review. 

In my opinion, what is missing here may be a comprehensible example that demonstrates the advances and limitations of each level of the theory. Can you show the advance achieved by each development via an exemplary material system? Can you compare the model’s description with any experimental understanding of the existing system? In my view this is what is missing in this review, severely shrinking the appeal of this novel development. 

Or, a series of schematics figures may be useful, too. Figure 1 is a great schematics for a model, and you can also draw the level of theory and key variables into a couple of more new figures.

I convince having this example-based narrative will broaden its readership for the journal Entropy. Without it, still this will be a nice set of literature review and I would still recommend its acceptance after only minor review, although the appeal to broader readership may be reduced. 

Author Response

We would like to thank the reviewer for their constructive comments.

At this point this review includes only the formalism we have developed.

The applications of the model for real systems will come in a future publication and we are still working on its implementation, both in equilibrium and in non-equilibrium cases.

A simple model calculation was already published in our previous work, which we can reproduce here for clarity.

As for work of others, due to copyright issues, we cannot reproduce their figures here.

In the revised version, we will put more emphasis on the novelty of this approach and its differences with existing methods. 

Sincerely,

Keivan Esfarjani

Reviewer 2 Report

This review paper (entropy-1978191/Esfarjani) presents a dense review of rigorous theoretical calculations for temperature-dependent crystal structures, lattice vibrations, and non-equilibrium transport. This work gives relatively detailed derivation of various equations of motion and Green’s function methods for determining anharmonic self-consistent phonons (including lattice expansion and stresses) and anharmonic transport in systems connected by leads obeying Langevin thermostats. This is a compilation of the authors’ previous work. As a review, I find it somewhat strange to focus solely on the authors’ own contributions, but perhaps this is the point of this work, solicited or not. If this is the purpose of this work, then I recommend its publication in Entropy and offer a few comments to perhaps improve and clarify parts of the manuscript:

(1) There are some typos and thus the manuscript should be carefully scoured. Few examples: missing parenthesis on pg. 1; commas missing around ‘however’ on pg. 1; question mark missing for question on pg. 2; misspelled ‘equilibrium’ on pg. 7; etc.

(2) Would be good to more generally cite relevant literature (yours and own) when going through all the various derivations.

(3) Define all parameters when they are introduced in the various equations. There are many instances where they are not, and many where they are. For example, p_i, m_i, and Greek symbols in Eq. 3.

(4) Shouldn’t Eq. 6 also have sums over k’ and lambda’, that is, why are the two y variables not independent? Does this have something to do with the creation and annihilation operators? Perhaps a bit more explanation here.

(5) Bit strange not to also label the equations that occur between Eq. 15 and Eq. 16.

Author Response

We thank the referee for their constructive comments.

Yes, this work is not a general review, but mainly a review of our own contribution to the development of lattice dynamical theories of anharmonic materials. But for the sake of completeness, we have given credit to similar works by other groups. In the revised version, we will more precisely clarify the differences between the present and previous contributions by others.

Typos are fixed, thanks!

Indeed some equations did not have number on them. The reason was that they were probably intermediate steps in the derivations and were not referred to. 

Most derivations are either our own, or textbook approaches to which we did not refer as they were pretty much standard (such as the fluctuation-dissipation theorem, or the relationship between displacements and phonon coordinates). The less common formulas (such as DFN) usually have a reference. But for completeness, we will add in the revised version reference to standard formulas as well. 

Reviewer 3 Report

This paper gives a review on the theory of in and out of equilibrium lattice dynamics. The theoretical investigation on the out-of-equilibrium lattice dynamics is very timely given that recent growing attention to heat flow across a junction or an interface, where mounting evidence has shown that close to an interface, phonon distribution is nonequilibrium. Although the paper is well organized and clearly presented, there are a few minor concerns the authors could consider:

1.     It would be nice to have one or two numerical comparisons between the current and past treatments across an interface even in 1D system.

2.     How are the non-equilibrium averages discussed in the review affected by the device size and temperature differences between the leads? Some numerical demonstration would be helpful for the readers to appreciate the current treatment.

3.     There are several approximations throughout the derivation. As the authors mentioned in the conclusion, it would be desirable to compare the current method with the MD simulations to validate these approximations. One particular concern I have is over the decoupling approximation around Eq. 45. Could the authors give some justification over this approximation? Why one can approximate the unknown (out-of-equilibrium) distribution this way?

Author Response

We thank the reviewer for the constructive comments.

1-The numerical application of the present formalism will appear in a future publication. But the main result is that, to leading order, one can still use the harmonic result for the transmission and Green's function except that they must be calculated with the effective force constants.

2-The large non-equilibrium effect will happen for large temperature differences across the sample. Essentially, larger thermal expansion will occur near the hotter end, causing a larger change in the harmonic force constants near that boundary. As I said above, numerical illustrations will come in a future publication.

3- Indeed in our opinion the largest approximation beyond the NESCP is in the decoupling approximation of anharmonic force autocorrelation <AA>. This is inspired from Wick's theorem which is exact and works in equilibrium. In the absence of any better scheme, we have used wick's theorem as an approximation, which should work well at least for smaller temperature differences. This term which has 4th or higher powers of displacements would be small if atoms involved are different. So the decoupling is justified in this case.  The major neglected contribution in <AA> is when all 4 atoms are identical. The decoupling will catch a portion of it, but it is unclear how large the remaining contribution is. This can be assessed by doing a comparison with MD simulation using the same force field. We plan to perform this simulation in a future work as this paper is mainly concerned with the formalism.   

Round 2

Reviewer 1 Report

Dear Authors, 

Thank you for the effort revising the manuscript.

I still believe there is a room for improvement, to clearly show the original insight that you offer through the review. However, to have a comprehensive collection of your work in a single short review may be a good thing. I would consent publication of the revised version.

I look forward to your follow up publication after this. Thank you.

Reviewer 2 Report

The manuscript is now fine in its present form. Publish at will.

Reviewer 3 Report

The authors have addressed the concerns from all the three reviewers. Therefore I would recommend for publication.